# Different antigenic distance metrics generate similar predictions of influenza vaccine response breadth despite moderate correlation

W. Zane Billings[1,2]*, Yang Ge[1], Amanda L. Skarlupka[3], Savannah L. Miller[1,2],
Hayley Hemme[1,2], Murphy John[1,2], Natalie E. Dean[4], Sarah Cobey[5],
Benjamin J. Cowling[6], Ye Shen[1], Ted M. Ross[7,8], Andreas Handel[1,2]*

**1** Department of Epidemiology and Biostatistics, University of Georgia, Athens, Georgia, United States of America, **2** Center for the Ecology of Infectious Diseases, University of Georgia, Athens, Georgia, United States of America, **3** Vivli, Cambridge, Massachusetts, United States of America, **4** Department of Biostatistics and Bioinformatics, Emory University, Atlanta, Georgia, United States of America, **5** Department of Ecology and Evolution, University of Chicago, Chicago, Illinois, United States of America, **6** Division of Epidemiology and Biostatistics, Li Ka Shing Faculty of Medicine, University of Hong Kong, Hong Kong, Hong Kong, **7** Center for Vaccines and Immunology, Department of Veterinary Medicine, University of Georgia, Athens, Georgia, United States of America, **8** Florida Research & Innovation Center, Cleveland Clinic, Port St. Lucie, Florida, United States of America

* ahandel@uga.edu (AH); wesley.billings@uga.edu (WZB)

## Abstract

### Introduction

Influenza continuously evolves to escape population immunity, which makes formulating a vaccine challenging. Antigenic differences between vaccine strains and circulating strains can affect vaccine effectiveness (VE). Quantifying the antigenic difference between vaccine strains and circulating strains can aid interpretation of VE, and several antigenic distance metrics have been discussed in the literature. Here, we compare how the predicted breadth of vaccine-induced antibody response varies when different metrics are used to calculate antigenic distance.

### Methods

We analyzed data from a seasonal influenza vaccine cohort that collected serum samples from 2013/14 – 2017/18 at three study sites. The data include pre- and post-vaccination HAI titers to the vaccine strains and a panel of heterologous strains. We used that data to calculate four different antigenic distance measures between assay strains and vaccine strains: difference in year of isolation (temporal), *p*-Epitope (sequence), Grantham's distance (biophysical), and antigenic cartography distance (serological). We analyzed agreement between the four metrics using Spearman's correlation and intraclass correlation. We then fit Bayesian generalized additive mixed-effects models to predict the effect of antigenic distance on post-vaccination titer after controlling for confounders and analyzed the pairwise difference in predictions between metrics.

**Data availability statement:** Our dataset and code are archived on GitHub (https://github.com/ahgroup/billings-comp-agdist-public) and Zenodo (https://doi.org/10.5281/zenodo.15522148).

**Funding:** The following authors received partial funding for this work. NED received partial funding from NIH contract(s)/grant(s) R01-AI139761. TMR received partial funding from the Georgia Research Alliance as an Eminent Scholar. AH received partial funding from NIH contract(s)/grant(s) U01AI150747, R01AI170116, and 75N93019C00052. YS received partial funding from NIH contract(s)/grant(s) R35GM146612, R01AI170116, and 75N93019C00052. SC received partial funding from NIH contract(s)/grant(s) R01AI170116. All other authors declare no funding for this work. The funders had no role in study design, data collection and analysis, decision to publish, or preparation of the manuscript.

**Competing interests:** I have read the journal's policy and the authors of this manuscript have the following competing interests: BJC has consulted for AstraZeneca, Fosun Pharma, GlaxoSmithKline, Haleon, Moderna, Novavax, Pfizer, Roche, and Sanofi Pasteur. None of these companies were involved in the formulation of the study or the decision to publish or conduct the study. All other authors declare no potential conflicts of interest.

## Results

The four antigenic distance metrics had low or moderate correlation for influenza subtypes A(H1N1), B/Victoria, and B/Yamagata. A(H3N2) distances were highly correlated. We found that after accounting for pre-vaccination titer, study site, and repeated measurements across individuals, the predicted post-vaccination titers conditional on antigenic distance and subtype were nearly identical across antigenic distance metrics, with A(H3N2) showing the only notable deviation between metrics, despite higher agreement for that subtype.

## Discussion

Despite moderate correlation among metrics, we found that different antigenic distance metrics generated similar predictions about breadth of vaccine response. Costly titer assays for antigenic cartography may not be needed when simpler sequence-based metrics suffice for quantifying vaccine breadth.

### Author summary

Influenza viruses change rapidly, so designing vaccines that remain effective is difficult. Small differences between the strains in the vaccine and strains in circulation can reduce protection. We can use a variety of methods to measure how "different" two strains are, but these methods can disagree.

We compared four ways of measuring these differences (genetic, biochemical, antigenic cartography, and time-based). Using immunological data from several flu seasons, we measured strain differences four ways. Then, we compared the relationship between immunogenicity and distance for each method. Our comparisons used a causal framework so we can identify valid conclusions from observational data.

We found that the four measures did not always agree with each other. But, the metrics produced similar predictions about the breadth of immune response to vaccination. Thus, complex and expensive laboratory tests may not always be necessary. Many studies could use simpler methods to save time and money. These results may aid in evaluation of future influenza vaccines

## Introduction

Influenza viruses constantly evolve over time. As host immunity induces selective pressure, new influenza strains accumulate mutations, a phenomenon called antigenic drift [1–6]. As mutations accumulate, antigenic drift leads to vaccine escape [7–9]. Seasonal influenza vaccines are formulated based on the strains that are expected to

circulate, but imperfect matches occur between selected vaccine strains and circulating strains in some years, and vaccine effectiveness (VE) varies annually [10]. A major determinant of VE is the similarity between vaccine strains and circulating influenza strains [11–20]. While previous studies have analyzed how mismatch between a circulating strain and the vaccine reduces VE, a full understanding of how viral changes affect vaccine response requires quantitative antigenic distance calculations [21–25]. If our goal is the development of a broadly-protective (or even "universal") influenza vaccine, which induces a robust immune response to both historical and future influenza strains, defining a broad response is a crucial first step. Defining a broad response relies on accurate measurements of antigenic distance.

The most common method for quantifying antigenic distance between influenza strains is antigenic cartography, which relies on extensive serological data generated against a wide panel of strains [26]. Briefly, statistical dimension reduction techniques are used to reduce large panels of serological data to fewer dimensions, and pairwise distances are calculated between strains in the reduced space. Serum samples from many individuals with wide assay panels are necessary to create stable cartographic maps. Cartographic distance has proven useful in understanding influenza evolution, but validating the ability of cartography to estimate population-level protection is difficult because of the required data [27–29]. Sequence-based methods can accurately predict cartographic distance based on genetic sequences of influenza strains, but still rely on accurate serological data for calibration [30–38]. Furthermore, multiple cartography methods yield different maps on the same data [26,27,39–41]. Maps based on HAI titers also incorporate bias from HAI assays, which are often not replicable between labs [42,43] and do not always accurately reflect differences in common antigenic phenotypes, also called antigenic clusters [21–25,35,39,44–46]. While cartographies can be generated from alternative assays [47–49], HAI is still the most common immunological assay used for influenza and the majority of highly-cited cartographies in use are based on HAI [26–28,50,51].

We can also compute antigenic distance without serological data. Simpler antigenic distance metrics calculated from genetic or amino acid sequences correlate with vaccine effectiveness at a population level [52–54], even though they only weakly correlate with antigenic distances derived from serological data [27,32,55]. Influenza strains that evolve to escape prior immune response typically have mutations at the same important genetic sites [56–58], and advanced predictive models consistently identify properties of the amino acid sequence of the major antigens as important predictors of vaccine escape [59–62]. Analyses of vaccine response or immunogenicity based on temporal [63–68] or sequence-based distances can provide information about the breadth of the vaccine response [30,44,52–55,69–71]. Taken together, these results imply that genetic analyses should provide important information about antigenic evolution without the need for serology. A direct comparison of antigenic distance methods is necessary to determine whether serological and sequence-based antigenic distance calculations can provide the same information in a practical setting. Specifically, we compare temporal distance (difference in the years of strain isolation), p-Epitope sequence distance [52], Grantham's sequence distance [72], and cartographic distance.

To compare the implications of multiple antigenic distance metrics on practical outcomes, we perform a secondary data analysis of an influenza vaccine cohort with a panel of HAI measurements to historical strains for each individual. We aim to assess whether low-cost measurements of antigenic distance between the vaccine strain and circulating strain may be similarly informative of the post-vaccination immune response. We find that, despite the modest correlation in antigenic distance metrics, these different metrics lead to similar conclusions about vaccine response to antigenically distant strains. Our results suggest that implementing costly antigenic analyses may not be necessary, as simple sequence-based measures lead to similar predictions about vaccine response as antigenic distance varies.

## Methods

### Study ethics

Study participants in the UGAFluVac study were enrolled into the study with written informed consent at their respective study site. The study procedures, informed consent, and data collection documents were previously reviewed by the University of Georgia Institutional Review Board (IRB), and by WCG IRB. We only used deidentified data from UGAFluVac, and our study was determined to be not human research and exempt from review by the University of Georgia IRB.

## Data source

**Immunological data.** The data for our study are from a human vaccination cohort study that has been described in detail previously [73–75]. Briefly, the study recruited participants at three study sites. The first two sites were Pittsburgh, PA, USA (PA site), and Port St. Lucie, FL, USA (FL site), beginning in the 2013/14 influenza season (approximately September through March [76]) and continuing through the 2016/17 influenza season. In January 2017, the study moved to Athens, GA, USA (UGA site). Participants visited the study site at least two times. At the first visit, patients completed a demographic questionnaire, gave a pre-vaccination serum sample, and received a Fluzone (Sanofi-Pasteur) seasonal influenza vaccine. At a follow-up visit approximately 21 days after the first visit, individuals returned and donated a post-vaccination serum sample. Individuals under 65 years of age received a standard dose Fluzone vaccination, and individuals aged 65 and older were given the choice between standard dose (SD) and high dose (HD) Fluzone vaccines. The study was a prospective, open cohort design where individuals could enroll in multiple years in the study, but were not required to re-enroll in every consecutive year.

Researchers tested the pre- and post-vaccination serum samples with a panel of hemagglutination inhibition (HAI) assays to the homologous strains (the strains included in the seasonal vaccine formulation), and a panel of historical, heterologous influenza virus strains. HAI assays are a common measurement for the strength of the antibody response, and correlate with the amount of antibodies in a serum sample that bind to the receptor-binding domain of the influenza hemagglutinin protein [77,78]. Strains included in the historical panel represented major lineages of circulating influenza viruses. See the Supplement for details on the Fluzone vaccine formulation and for a list of strains used in each season.

Each HAI assay in our dataset can be defined by its (1) subtype, (2) vaccine strain, and (3) assay strain. The broadest grouping is "subtype", which we use to describe both influenza A subtypes (H1N1 and H3N2) and influenza B lineages (Pre-divergence or Pre-split, Victoria-like and Yamagata-like). The vaccine strains associated with an HAI assay are the strains used in the Fluzone vaccine formulation in the season when the serum sample was collected. Each assay has three or four associated vaccine strains, depending on whether the individual who gave the serum sample received a trivalent or quadrivalent vaccine (see Supplement for details on the vaccine formulations). Finally, the assay strain for a given HAI assay is the strain of the actual virus added to the serum sample during the assay. We only compared vaccine strains and assay strains of the same subtype in our analysis.

For our secondary data analysis, we extracted previously deidentified records from the 2013/14–2017/18 influenza seasons. The study is ongoing and more assays are available, but the size of the historical panel was reduced after the 2017/18 season, and there would not be enough heterologous strains to estimate stable cartographic maps, so we limit our analysis to these seasons of data. Since examining the effect of vaccine dose was not our main focus here, and we previously observed dose-dependent differences in the heterologous response [79], we only included individuals who received SD vaccines in our study. We included all participants from the specified seasons who received SD vaccine and had records for both pre-vaccination and post-vaccination serum samples in our analysis. Our primary outcome of interest was the post-vaccination HAI titer (the reciprocal of the highest serum dilution that shows agglutination), which we log transformed:

$$\text{transformed titer} = \log_2 \left( \frac{\text{HAI titer}}{5} \right).$$

Our final dataset for analysis contained one pair of transformed titers (pre- and post- vaccination) per person-year per assay strain in the historical panel, along with corresponding covariate measurements.

We divided the titer by 5 before taking the log because the HAI assay had a lower limit of detection (LoD) of 10, and an upper LoD of 20,480. Values below the LoD were coded as titers of 5 in the dataset, corresponding to a transformed titer of 0. All observed values in our dataset were below the upper LoD. We used the same outcome definitions defined in our previous work on this dataset [79].

**Sequence data.** We computed the pairwise antigenic distance for all strains used in the dataset (see the Supplement for a complete list). We used four different methods to compute the antigenic distance: temporal distance, dominant *p*-Epitope distance [52], Grantham's distance [72], and cartographic distance [26]. We calculated the temporal difference as the difference in the year of isolation between two strains (we had no assay strains with years of isolation subsequent to the vaccine strain, so all distances are positive). The dominant *p*-Epitope distance is the maximum of the Hamming distances [80] calculated for each of the five major epitope sites on the hemagglutinin head. Grantham's distance is similar to the Hamming distance on the entire HA sequence, but weights each substitution between strains by a score that is larger for amino acids with very different biochemical or biophysical properties. Finally, we conducted antigenic cartography using Racmacs [81] and reduced all of our cartographic maps to two dimensions. For complete details on antigenic distance calculation, see the Supplement.

To calculate the sequence-based, distances, we obtained sequences for the HA amino acid sequences for each of the strains used in the UGAFluVac data from either the U.S. National Center for Biotechnology Information (NCBI)'s GenBank database [82,83], the UniProt dataset [84], or GISAID's EpiFlu database [85,86]. Accession numbers and sources for the sequence for each strain are shown in the supplement.

## Statistical analyses

We first summarized demographic information about the cohort in a descriptive analysis, stratifying by measurements, individuals, and person-years to demonstrate the multilevel structure of our data.

We calculated reliability statistics between the different antigenic distance metrics, using antigenic distances for all pairs of vaccine strains and assay strains that were present in the study design (instead of examining the reliability between all strains pairwise). As an omnibus test of measurement reliability, we calculated the intraclass correlation (ICC) using a Bayesian two-way mixed effects model for consistency and a single score, i.e., ICC(3,1) in the Shrout-Fleiss taxonomy [87–89]. The Supplement shows the exact model we fit and formula for calculating the ICC. To analyze which metrics drove disagreement or agreement, we also calculated the Spearman rank correlation coefficient [90] between each pair of antigenic distance metrics. We show credible intervals for the Spearman correlations in the Supplement, calculated with the Bayesian bootstrap [91].

We built generalized additive mixed-effects models (GAMMs) and linear mixed-effects models (LMMs) with the transformed post-vaccination titers as the outcome, [92,93] and adjusted for interval censoring [94] (see Supplement for details). To answer our primary question, we modeled antigenic distance in two ways. For the LMM, we included a linear effect of antigenic distance that was allowed to vary by subtype. For the GAMM, we modeled antigenic distance using a flexible semi-parametric spline that allows the relationship to be nonlinear, but constrained. We adjusted for effects of birth year, age, sex, race/ethnicity, effects of the vaccine and assay strain, differences between study sites, and repeated measurements from the same individual.

We fit the models in a Bayesian framework using weakly informative priors chosen by a prior predictive simulation [92,95]. We obtained posterior samples of the model parameters using the No U-Turn Sampler (NUTS) algorithm implemented by Stan [96,97], via the brms [98–100] and cmdstanr [101] packages for R [102]. After obtaining the posterior samples, we calculated marginal posterior predictions for interpolated values of the normalized antigenic distance [103]. We summarized the posterior prediction samples with a mean point estimate and 95% highest density continuous interval (HDCI). We compared the GAMM and LMM for each antigenic distance metric using the leave-one-out expected log pointwise predictive density (LOO-ELPD) that is conceptually similar to model selection using cross-validation in a frequentist scenario [104,105,106]. See the Supplement for extensive details on our models.

To examine the differences in predictions across each of the antigenic distance metrics, we compared the slope and intercept for LMMs and the fold change in predicted post-vaccination HAI titer for the LMM and GAMM since the GAMM has no equivalent simple parametrization (fold change comparisons are shown in the Supplement). We extracted the fixed effects coefficients from the models, along with the random effects and residual variance components. We computed the

variance contribution of the fixed effects [107] and calculated the proportion of variance explained by each of the variance components, defining the total variance as the sum of the residual variance parameter, the fixed effects variance contribution, and all random effects variance components.

## Implementation

We conducted our analysis with R version 4.4.1 (2024-06-14 ucrt) [102] in RStudio version 2024.09.0+375 [108]. Our analysis pipeline was implemented in targets [109]. We used the packages here [110], renv [111], and the tidyverse [112] suite for data curation and project management and the packages marginaleffects [103], tidybayes [113], ggdist [114,115], bayesboot [116], and loo [104,105,116] for formal analysis. We used the packages ggplot2 [117] and GGally [118] for generating figures; and the packages gtsummary [119] and flextable [120] for generating tables. We generated the manuscript using Quarto version 1.6.40 [121] along with the R packages knitr [122–124] and softbib [125]. We implemented our Bayesian models with the brms package [98–100] using the cmdstanr backend and cmdstan version 2.34.1 [101] as the interface to the Stan programming language for Bayesian modeling. The Supplement contains more exhaustive details on our methodology, including instructions for reproducing our results. Our dataset and code are archived on GitHub (https://github.com/ahgroup/billings-comp-agdist-public) and Zenodo (DOI: 10.5281/zenodo.15522148).

## Results

### Data description

Our dataset included 54,101 pairs of pre-vaccination and post-vaccination HAI titer measurements drawn from 677 individuals who contributed 1,163 person-years to the study across three different study sites. The contributions of paired measurements, person-years, and unique participants from each study site are shown in Table 1. In a given year, each individual contributed three (trivalent vaccine in 2013/14 and 14/15) or four (quadrivalent vaccine from 2015/16 onward) homologous HAI assay pairs, along with a number of heterologous assay pairs, which varied by season due to the change

**Table 1.** Counts of HAI assay pairs, person-years, and new participants who enrolled for the first time that season contributed by each study site for the duration of the study. The PA and FL study sites operated from September 2013 to December 2016 and the GA study site began operating in January 2017 (during the 2016/17 influenza season).

| | Season | | | | | |
|---|---|---|---|---|---|---|
| | 2013/14 | 2014/15 | 2015/16 | 2016/17 | 2017/18 | Total |
| Paired HAI assays, n | | | | | | |
| FL | 2459 | 6597 | 6656 | 6188 | 0 | 21900 |
| PA | 2163 | 3716 | 4131 | 3136 | 0 | 13146 |
| UGA | 0 | 0 | 0 | 6815 | 12240 | 19055 |
| Overall | 4622 | 10313 | 10787 | 16139 | 12240 | 54101 |
| Person years, n | | | | | | |
| FL | 60 | 150 | 128 | 119 | 0 | 457 |
| PA | 73 | 88 | 81 | 64 | 0 | 306 |
| UGA | 0 | 0 | 0 | 145 | 255 | 400 |
| Overall | 133 | 238 | 209 | 328 | 255 | 1163 |
| New participants, n | | | | | | |
| FL | 60 | 113 | 37 | 31 | 0 | 241 |
| PA | 73 | 46 | 2 | 12 | 0 | 133 |
| UGA | 0 | 0 | 0 | 145 | 158 | 303 |
| Overall | 133 | 159 | 39 | 188 | 158 | 677 |

in historical panels each year, and by individual due to random lab and assay issues. Each person-year represented in the data contributed a median of 48 HAI assay pairs (range: 8–52 pairs). Additional demographic information about our cohort is provided in the Supplement (summaries of race/ethnicity, sex assigned at birth, contributed person-years, age at enrollment, and pre-vaccination titer).

**Antigenic distance metrics have low or moderate correlation for all subtypes except A(H3N2)**

First, we examined the overall agreement between the different distance metrics. We analyzed agreement using the intra-class correlation (ICC), shown in Table 2. ICC was low for all subtypes except A(H3N2), and the credible interval included zero for all subtypes except A(H3N2), so despite the moderate point estimate for B/Yamagata with a high upper limit, there was low consistency in antigenic distance measurements across methods. For A(H3N2), we observed moderate agreement across methods. Our ICC results indicate for each subtype except A(H3N2), at least one of the antigenic distance metrics systematically disagrees from the other.

To better understand the lack of overall agreement, we computed the Spearman rank correlation between each pair of metrics (again, separately for each subtype). Fig 1 shows the pairwise scatterplots and correlation coefficients. The pairwise correlations between distance measurements varied widely across subtypes and combinations, indicating that low agreement was not driven by a specific metric or subtype. All distance metrics tended to correlate well for H3N2. Distance metrics correlated highly for both influenza B lineages with the exception of the cartographic distance, which had a moderately high correlation with the other three distances for B/Yamagata and a low correlation with the other three distances for B/Victoria. The only high correlation for A(H1N1) was between Grantham and *p*-Epitope distance, with small correlations between the other distance metrics. Grantham and *p*-Epitope distances correlated well for all strains (although it was notably lower for A(H1N1)), which we expected given the similarity between the measures. The Supplement contains a table with credible intervals for each correlation.

**Predicted vaccine response breadth is similar across antigenic distance metrics, despite the low between-metric correlation**

Examining the agreement and pairwise correlations between the different distance metrics is useful for understanding which metrics disagree most, but these disagreements do not necessarily translate into different predictions about vaccine response. We built LMMs and GAMMs to model the effect of antigenic distance after controlling for multiple host and assay features.

To quantify whether the effect of antigenic distance deviated strongly from a linear effect, we calculated the LOO-ELPD for the GAMM and LMM models fit with each antigenic distance metric, shown in Table 3. LOO-ELPD is comparable to frequentist information metrics such as the Akaike Information Criterion (AIC), and differences in ELPD strongly supported the linear model for every antigenic distance metric. The ratio of the difference in ELPD was always much greater than its

**Table 2. Intraclass correlation (ICC) across all antigenic distance measurements, calculated separately for each subtype or lineage (strain type). The posterior distribution for each ICC was calculated as the ratio of variance components for vaccine strain and assay strain divided by the sum of all variance components, estimated with a Bayesian model. Numbers shown are the mean and 95% highest density credible interval (HDCI) of the posterior distribution of ICCs.**

| Strain Type | ICC |
|---|---|
| H1N1 | 0.09 (0.00, 0.24) |
| H3N2 | 0.35 (0.20, 0.53) |
| B-Yam | 0.21 (0.00, 0.42) |
| B-Vic | 0.03 (0.00, 0.12) |

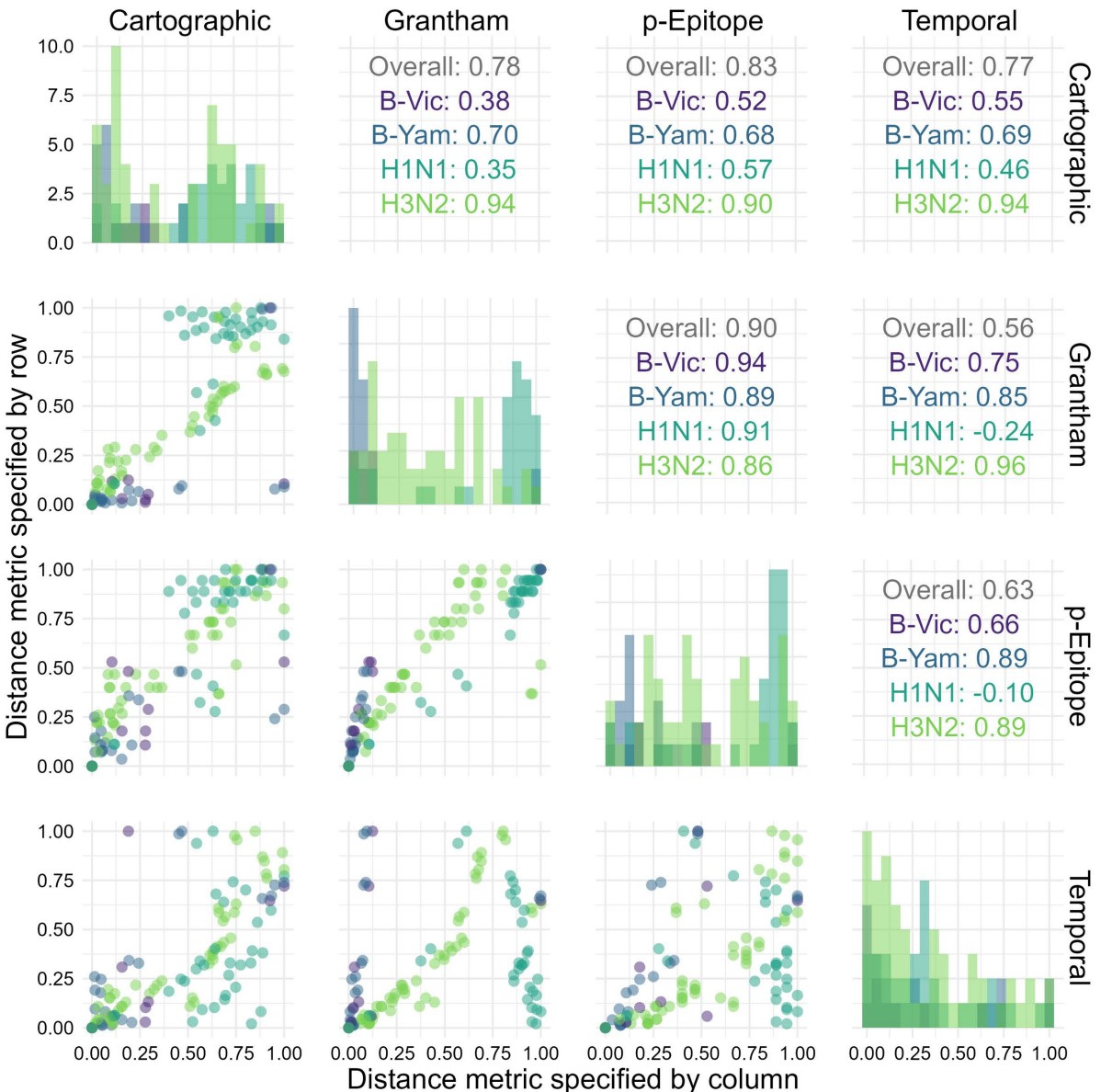

**Fig 1. Distribution and correlation plots for each of the antigenic distance metrics.** For each HAI assay in the dataset, we calculated the antigenic distance between the vaccine and assay strains with four different methods. We examined the distribution (shown along the diagonal) and the correlation between the different metrics for the same pairwise comparisons (we show pairwise scatterplots in the plots below the diagonal, and overall Spearman's correlation values in the plots above the diagonal). We include each unique combination as only one point in this plot. We calculated correlation coefficients separately for each subtype – colors in the plot indicate subtype.

standard error, so the difference between models can be trusted for model selection. Including spline terms to account for nonlinearity did not improve the model fit.

Fig 2 shows how the average post-vaccination titer predicted by the model changes along with antigenic distance for each subtype. For both influenza B lineages, the data were sparsely measured across the span of any of the antigenic distance metrics, making the GAMM predictions difficult to distinguish from the LMMs. Both influenza A subtypes showed a larger difference in predictions made by the GAMMs vs. the LMMs where the GAMMs predicted non-monotone

**Table 3. Differences in expected log pointwise predictive density (ELPD) from the best-fitting model, estimated by the leave-one-out (LOO) method for all models and all antigenic distance metrics.** We fit the models separately for each antigenic distance metric, so comparisons are shown separately. The ΔELPD is the difference in ELPD between the LMM and the GAMM, so a positive number indicates the LMM performed better than the GAMM, and a larger number means the LMM outperforms the GAMM more. We show the ΔELPD±its standard error, along with the ratio of the estimate to its standard error.

| Metric | ΔELPD (LMM - GAMM) | ΔELPD/ SE |
|---|---|---|
| Cartographic | 108.14 (±19.96) | 5.4 |
| Grantham | 203.64 (±24.23) | 8.4 |
| Temporal | 47.16 (±11.17) | 4.2 |
| p-Epitope | 290.25 (±35.48) | 8.2 |

relationships between post-vaccination titer and antigenic distance. The LMM and GAMM were most similar for cartographic distance for both A(H1N1) and A(H3N2), perhaps suggesting that cartographic distance partially accounts for nonlinear effects of antigenic distance. There were some interesting trends in the shape of the spline curves, but the nonlinear effects for the *p*-epitope and Grantham distance did not appear to match the data well. Combined with the lack of ELPD support (Table 3), the spline models are likely picking up random fluctuations that may be partially driven by gaps in antigenic distance space rather than by true non-monotone signals (see the Supplement for an analysis of the gaps in antigenic distance space).

Since the linear model had better ELPD support for all metrics (Table 3), we focused on attempting to understand the effects in the linear model. Other than the normalized antigenic distance effect, the other effects were similar across the four models (what we expect). Table 4 shows the estimated fixed effects coefficients from our models. The effects of sex and race/ethnicity were negligible, and the effects of age and birth year appear to be highly negative because they are not identifiable in our dataset, but together they provide a non-negligible contribution for each individual. Log pre-vaccination titer had a strong positive effect on post-vaccination titer as expected. We did not interpret those effects further, since we did not control for potential confounders of relationships other than the effect of antigenic distance on the outcome. The effect of antigenic distance was negative for all four models, as we would expect, but the magnitude of the effect varied. While the point estimates were similar, the effect size for *p*-epitope was the smallest and the effect size for cartographic distance was the largest. The effect size for the cartographic distance also had the most density away from zero. Only the temporal distance model had an HDCI for the distance effect that included zero.

We also attempted to understand the variance contributions in the model by decomposing the variance (Table 5). The fixed effects explained the most variance of the three model components in all four models. The contribution of the residual variance was nearly identical in all four models, suggesting that the random effects are more important in some models than others, without explaining any additional variance. The variance explained by the assay strain, vaccine strain, study site, and subject variance components was similar across the four models, with the most noticeably different contribution being the effect of the subtype. The subtype apparently explained more variance in the temporal and Grantham distance models than in the cartographic and *p*-Epitope distance models, suggesting that those metrics might be more affected by differences in subtypes. Overall, the fixed effects were typically slightly more important than the random effects, but the variance explained by the random effects was still large for each model.

## Predictions made by different antigenic distance metrics are similar after accounting for host factors

Finally, we directly compared estimates from the models across normalized antigenic distance metrics for each subtype (Fig 3). Since the LMM is easier to interpret and was supported by our ELPD analysis, we examined the slope and intercept for each subtype across the four antigenic distance metrics. The intercepts (representing the predicted

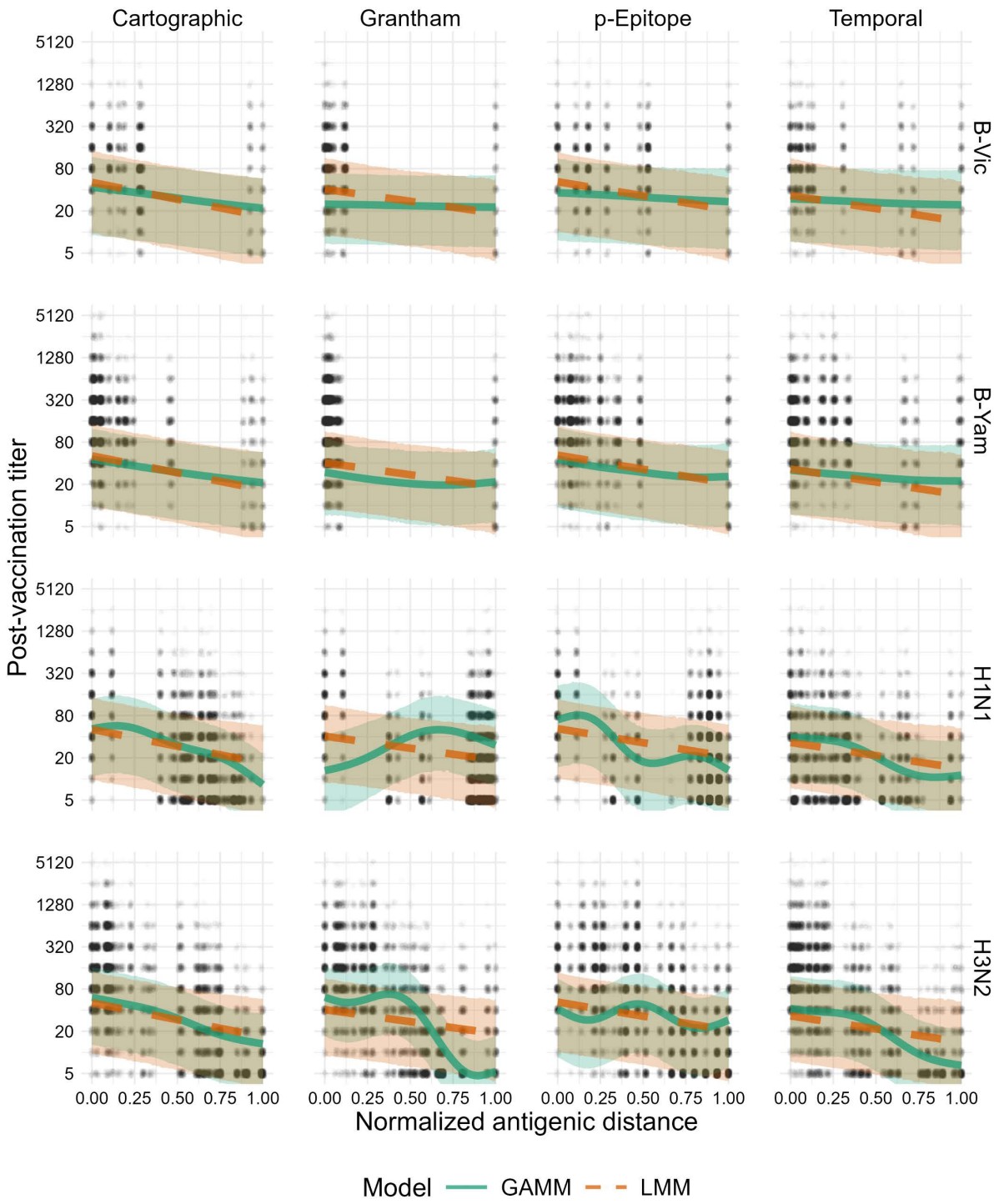

**Fig 2. Model predictions for both the GAMM and LMM.** Solid green lines and green ribbons show the mean and 95% highest density continuous interval (HDCI) for GAMM predictions. Dashed orange lines and orange ribbons show the mean and 95% HDCI for LMM predictions. Circular points show the data values. Each subplot shows the model predictions for a particular subtype (changes by row) and distance metric (changes by column). Outcomes shown on the plot are predicted post-vaccination titers for an average individual to an average strain (see Supplement for computational details).

**Table 4. Coefficients for all of the fixed effects included in our primary models. The model coefficients for scaled birth year, scaled age, sex (effect of being male relative to being female), race/ethnicity (effect of being non-Hispanic white or Caucasian vs. any other self-reported identity), log pre-vaccination titer, and normalized antigenic distance. We fit a separate model for each of the metrics, but the variables are standardized the same way across all four models so the coefficients are on the same scale across all models.**

| Metric | Birth Year | Age | Sex[1] | Race/Ethnicity[2] | Log pre-vaccination HAI titer | Normalized antigenic distance |
|---|---|---|---|---|---|---|
| Cartographic | -3.14 (-4.08,-2.20) | -3.46 (-4.37,-2.55) | 0.01 (-0.05, 0.06) | 0.03 (-0.02, 0.08) | 0.78 (0.77, 0.79) | -1.61 (-2.42,-0.58) |
| Grantham | -3.19 (-4.12,-2.26) | -3.50 (-4.41,-2.60) | 0.01 (-0.05, 0.07) | 0.03 (-0.02, 0.09) | 0.78 (0.78, 0.79) | -1.14 (-2.03,-0.02) |
| p-Epitope | -3.13 (-4.08,-2.19) | -3.45 (-4.36,-2.53) | 0.01 (-0.05, 0.06) | 0.03 (-0.02, 0.08) | 0.78 (0.78, 0.79) | -1.33 (-2.12,-0.50) |
| Temporal | -3.17 (-4.11,-2.24) | -3.48 (-4.41,-2.58) | 0.01 (-0.05, 0.06) | 0.03 (-0.02, 0.08) | 0.78 (0.78, 0.79) | -1.24 (-2.38, 0.16) |

1 Reference: Male (vs. female)

2 Reference: Non-Hispanic white (vs. other)

**Table 5. Variance contributions to the total variance estimated in the model. To estimate the fixed effects variance contribution as the variance of the estimated linear predictor, while the residual variance and random effects variance contributions (all variance contributions other than the fixed effects and residual variance) are estimated as model parameters. All contributions are rounded to the nearest percent and may not sum (rowwise) to 100 due to rounding error.**

| Metric | Residual variance | Fixed effects | Total random effects | Specific random effects | | | | |
|---|---|---|---|---|---|---|---|---|
| | | | | Subtype | Assay strain | Vaccine strain | Study site | Subject |
| Cartographic | 12% (9, 15) | 50% (38, 62) | 36% (22, 52) | 13% (4, 26) | 1% (1, 2) | 3% (1, 5) | 11% (0, 31) | 6% (4, 7) |
| p-Epitope | 12% (9, 15) | 50% (38, 60) | 36% (24, 51) | 11% (3, 24) | 5% (3, 6) | 3% (1, 5) | 10% (0, 28) | 6% (4, 7) |
| Grantham | 11% (9, 14) | 48% (37, 58) | 40% (27, 53) | 17% (5, 33) | 4% (2, 5) | 3% (1, 7) | 8% (0, 25) | 6% (4, 7) |
| Temporal | 11% (8, 13) | 44% (33, 54) | 44% (32, 57) | 23% (9, 40) | 4% (2, 5) | 3% (1, 7) | 7% (0, 23) | 5% (4, 6) |

post-vaccination titer to the homologous strain of the specified subtype for an individual with no pre-vaccination antibodies) were similar across all metrics regardless of the subtype. The slopes varied more, indicating that the antigenic distance had a stronger effect on predicted titer for some metrics and subtypes. For both B lineages, estimates of the slope were nearly identical across antigenic distance metrics. For A(H1N1), the cartographic distance model had a lower slope than the other three antigenic distance metrics, but the credible interval still overlapped with the credible interval for the temporal distance. For A(H3N2), the slope for the p-Epitope distance was much smaller than the other slopes (reflecting our results in Fig 2), despite the high correlation between the antigenic distances for A(H3N2) (Fig 1). We can only perform a visual inspection of these overlaps, because there is no existing approach to combine posterior distributions across the four models.

Furthermore, these estimates do not take variance from the random effects in our model into account. To analyze predictions for both the LMM and GAMM, with the random effects variances included in uncertainty calculations, we directly compared predictions from the models and saw much higher overlap (shown in the Supplement), as we would expect when we include all of the variance in the data.

We compare the relative LOO-ELPD for each model in Table 6. The models are fit to the same set of predictors and data points and the antigenic distances are normalized, so the ELPDs are on the same scale and we can directly compare them. We found that all of the models had very similar performances – while the ELPDs were different between the four models, each contrast was smaller than the SE for either ELPD. For example, while the cartographic model had an ELPD around 150 points lower than the p-Epitope model, the SE for both estimates was around 470, so we cannot assume that these contrasts are meaningful differences. All of the models appeared to fit the data equally well.

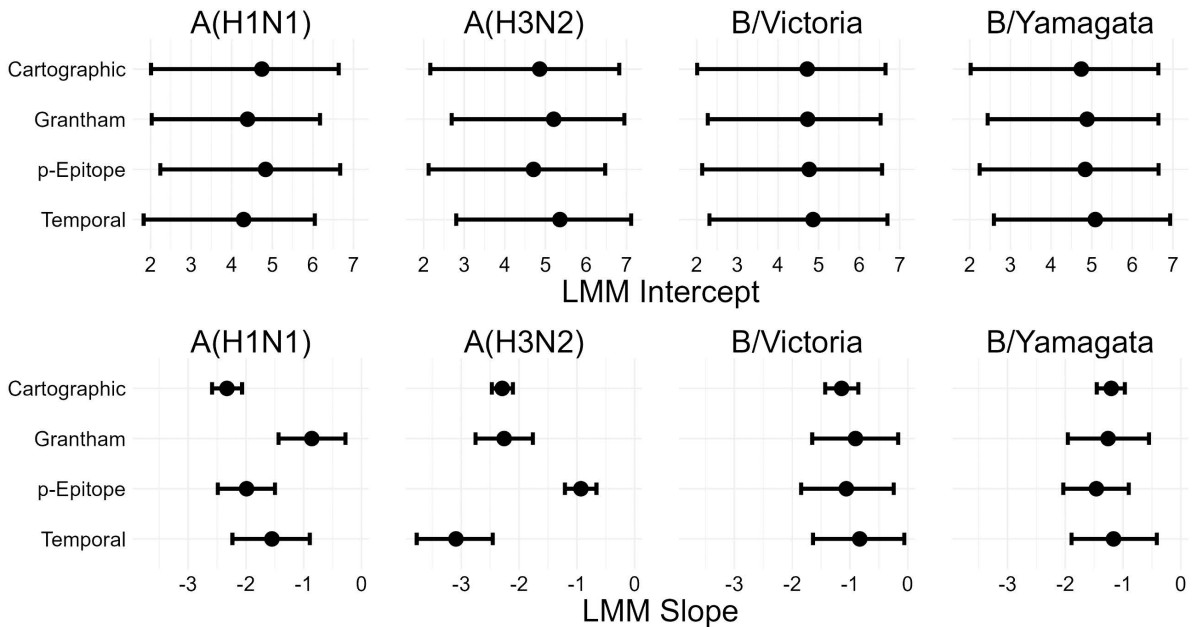

**Fig 3. Intercept and slope estimates stratified by subtype for each LMM (one for each distance metric).** Points and intervals show the mean and 95% HDCI of posterior samples of the indicated parameter. The top row of plots shows the mean and CI for estimates of the intercept, and the bottom row of plots shows the mean and CI for estimates of the slope. Columns of plots indicate which subtype the slope and intercept are for.

**Table 6. Expected log pointwise predictive density (ELPD) calculated for each of the linear mixed-effects models (LMMs) using the leave-one-out (LOO) method. For each metric, we show the estimated ELPD±its standard error. The differences between the model ELPDs were negligible.**

| Metric | LMM LOO-IC |
|---|---|
| Cartographic | 151131.5±471.6 |
| p-Epitope | 151188.6±471.7 |
| Grantham | 151250.6±472.1 |
| Temporal | 151188.4±472.2 |

## Discussion

We computed multiple antigenic distance metrics on the same set of influenza strains. Using immunogenicity data from a human cohort, we were able to compare cartographic data to sequence-based, biophysical, and temporal antigenic distance measures that have been used before for analyzing vaccine breadth. We then fit linear mixed-effects models (LMMs) and generalized additive mixed models (GAMMs) to the immunological data separately for each cohort, controlling for subtype, pre-vaccination titer, and multiple sources of random variation. By comparing the predictions and parameters from the estimated models across the four antigenic distance metrics, we were able to assess the similarity of the metrics in a more practical context.

We observed moderate correlations between the four antigenic distance measures for all subtypes except A(H3N2). Low ICC for influenza B could be due to the relative sparsity of the heterologous panel compared to the influenza A strains, but the four metrics clearly behave differently for A(H1N1) and A(H3N2) strains. Notably, the strains of A(H1N1) that have emerged since 1918 can be divided into two major groups — the group that is more similar to the 2009 or 1918 pandemic lineage, and the group that is more similar to the pre-2009 seasonal A(H1N1)

lineage. The temporal separation between the similar strains distorts the temporal distance considerably, and temporal distance cannot be used in a fair comparison due to this difference. While the low reliability observed between antigenic cartography and the sequence-based distances is harder to explain, we postulate that currently used genetic differences fail to adequately consider the indel mutation that differentiates the 2009 pandemic-like strains from the pre-2009 seasonal strains. Indel mutations are also important for distinguishing influenza B lineages, so incorporating a better gap penalty into genetic or biophysical distances may provide insight into the low reliability between metrics.

Despite the moderate correlations between metrics, we found that all four antigenic distance measures produced similar predictions about the heterologous vaccine response, regardless of subtype. Unexpectedly, the subtype generating the most different predictions was A(H3N2), which had the highest correlation between metrics. After we account for important confounders and other sources of variation, the differences between metrics seemed to disappear, with the exception of the unusually small slope for $p$-Epitope distance for influenza A(H3N2). Along with our pointwise prediction comparisons (shown in the supplement), these results suggest a systematic disagreement on the vaccine outcome scale between $p$-Epitope distance and other metrics for A(H3N2), which contrasts with the high pairwise correlations between $p$-Epitope and other metrics for this subtype. Perhaps important antigenic changes for H3N2 have occurred outside of the immunodominant epitopes, or features like glycosylation that might be more easily captured by Grantham or cartographic distance are important, but we were unable to identify specific strains driving this effect. Alternatively, the difference could be due to some form of noise or sampling error in our study — we have no data from equivalent human cohort studies with wide heterologous panels to compare our results to, so we do not know if this result is consistent.

Our overall results could imply that the differences between antigenic distance metrics can appear large but are small compared to between-subject and between-study variability in real life, or that accounting for interindividual differences or pre-vaccination titer helps to explain the differences between metrics. We also found that a linear model was sufficient for explaining the relationship between post-vaccination titer and antigenic distance, rather than a nonlinear model. For example, we might expect a tapering effect or a sharp drop-off, which could both be produced by the GAMMs. Notably, we even found that temporal distance tends to produce similar predictions to cartographic distance in this setting, despite the evidence for epochal antigenic evolution and emergence or circulation of multiple clusters in a single year [3,9,59,126]. Combining our the antigenic distance metrics we considered (and potentially other epidemiological or virological data) could produce a better with less nuisance variation across experimental units.

While we used data from a multicenter study with tens of thousands of measurements and over one thousand contributed person-years, our study still has some weaknesses. First, as a secondary data analysis, none of the data were designed with our questions in mind. While we have attempted to control for as much confounding as possible, we lack data on the exposure histories, including infections and prior vaccinations outside of the study, of individuals in our cohort that could confound our results [9,127]. While exposure history does not statistically confound the effect of antigenic distance in our study, exposure history could be a major source of between-subject variability. Controlling for exposure history would provide interesting conclusions in its own right but could help us resolve the effect of antigenic distance on immunogenicity.

Our results also only apply to the split-inactivated Fluzone standard dose vaccine. Higher doses can either help or hinder heterologous responses [128–130], and in a previous study we found that the heterologous antibody response varied by Fluzone vaccine dose [79], so our results might change for other vaccine doses or formulations. A balanced design with randomized vaccine design would be preferable for understanding the impact of vaccine design on agreement between antigenic distance metrics.

We also used cartographies based on our pre-immune human data, which were generated on the same data we analyzed. With access to multiple cartographies on the same data set or imputation techniques [131,132] we could treat

different cartographies as different antigenic distance metrics and compare cartographic distances in the same way. Our metrics also did not all cover antigenic distance space evenly as the strains in the historical panel were selected to cover a wide variety of years. However, there were several "gaps" between discrete antigenic distance values for A(H1N1) and the two B lineages, which could impact our estimates (see Supplement for details), and a broader panel with more evenly spaced strains would make our effect estimates more precise. Finally, we have no real proxy for the response to "future" strains. We could get a better predictive understanding of how the vaccine generates immune responses to future strains by testing serum samples from, say, 2016, to novel vaccine strains that have emerged since the samples were collected. Such measurements would allow us to validate the use of the historical panel as a proxy for future vaccine response. Longitudinal studies designed with long-term collection and multiplex assays in mind would be beneficial for answering similar questions about antigenic distance and vaccine breadth.

Overall, we found that simple antigenic distance metrics like Grantham's distance generated very similar predictions about vaccine breadth to distances based on antigenic cartography in our study. While some distance metrics potentially deviated, the effect was subtype specific (*p*-Epitope for A(H3N2) strains). While cartography is important for understanding the antigenic diversity and evolution of influenza, researchers analyzing vaccine breadth should not be afraid to use easier, potentially less biased metrics of antigenic distance.

## Supporting information

**S1 Table. Accession number and source for each HA sequence used in our analysis.** (XLSX)

**S2 Table. Strains used in the Fluzone standard dose vaccine formulation during each influenza season.** (XLSX)

**S3 Table. Heterologous strain panel used during each influenza season.** (XLSX)

**S4 Table. Full strain names and associated abbreviations for each strain used in the study.** (XLSX)

**S5 Table. Demographic characteristics of the study participants.** Summary statistics shown are count and column percent for sex, race, and contributed person-years; and median with range for age at first enrollment, birth year, and contributed HAI assays. Demographic variables were collected by a questionnaire from participants on the date they enrolled in a study season and received a vaccine. (XLSX)

**S6 Table. Prediction variance ratio across all antigenic distance measurements, calculated separately for each subtype or lineage (strain type).** The posterior distribution for each ratio was calculated as one minus the ratio of the prediction variance ignoring random effects to the prediction variance including random effects, estimated with a Bayesian model. Numbers shown are the mean and 95% highest density credible interval (HDCI) of the posterior distribution of variance ratios. (XLSX)

**S7 Table. Spearman correlation coefficients and 95% HDCIs estimated by Bayesian bootstrap for each influenza subtype.** Each pairwise comparison is shown only once to prevent confusion. (XLSX)

**S8 Table. Model diagnostics for the GAMMs and LMMs fit with each of the antigenic distance metrics.** We show the total number of divergences out of the number of samples along with other common diagnostic criteria. For each

model, we show the minimum ESS across all parameters, the minimum E-BFMI across chains, and the maximum R hat across all parameters.
(XLSX)

**S9 Table. Model diagnostics for samples from the prior distributions for our GAMMs and LMMs.** These samples are drawn only from the prior distributions and do not see the data. For each model, we show the minimum ESS across all parameters, the minimum E-BFMI across chains, and the maximum R hat across all parameters.
(XLSX)

**S10 Table. Diagnostics for the LOO-IC ELPD approximation.** Pareto $k$ is the primary diagnostic indicating whether the approximation is trustworthy and all Pareto $k$ values should be below 0.7. The $N_{eff}$ is the effective sample size and $R_{eff}$ is the ratio of the effective sample size to the true sample size – if there are too few effective samples relative to actual samples, we can get an optimistic evaluation of the approximation quality, but in general this matters less if the ESS is sufficiently high.
(XLSX)

**S11 Table. Pairwise Spearman rank correlations between antigenic distance values using the Grantham, FLU Substitution, and Hamming distance metrics.** We calculated correlations between two distances using the normalized distance values between every vaccine/assay strain pair for the given subtype. Numbers shown are the mean and 95% highest density continuous interval (HDCI) calculated by Bayesian bootstrapping.
(XLSX)

**S12 Table. Pairwise Spearman rank correlations between antigenic distance values using the Grantham, FLU Substitution, and Hamming distance metrics.** We calculated correlations between two distances using the normalized distance values between every vaccine/assay strain pair for the given subtype. Numbers shown are the mean and 95% highest density continuous interval (HDCI) calculated by Bayesian bootstrapping.
(XLSX)

**S1 Fig. The graphical causal model for our research question represented as a DAG.**
(PNG)

**S2 Fig. Pre-vaccination titers in our study to each of the assay strains.** The point shows the median and the line shows the IQR.
(TIF)

**S3 Fig. Post-vaccination titers in our study to each of the assay strains.** The point shows the median and the line shows the IQR.
(TIF)

**S4 Fig. Dispersion metrics for antigenic distance metrics.** A) Parallel coordinates plot showing how the estimated pairwise antigenic distances change for each of the antigenic distance metrics. Each line in the plot represents one vaccine strain and assay strain pair, and the connected points are the pairwise distance measured under each metric shown on the x-axis. When two lines cross, this indicates that two metrics assigned a different relative order to the pairwise combination. Note that Grantham and especially $p$-Epitope distances are integer-valued and concentrate measurements to specific points which potentially overlap (temporal distance is also integer valued but has enough spread to avoid a similar issue). B) The gap standard deviation (gap SD) for each subtype and antigenic distance metric. The posterior distribution of gap SDs was calculated using the bayesian bootstrap with reweighting. The red horizontal bar shows the mean of the bootstrap posterior and the error bars show the 95% highest density credible interval (HDCI).
(TIF)

**S5 Fig. Pairwise comparisons of predictions (from the LMMs) between each unique set of two metrics.** The y-axis shows the fold change in predictive titers between metrics, and the two metrics being compared in each subplot are shown as the subplot labels. Each line represents the predictions for the first metric in the pair at a given antigenic distance value divided by the predictions for the second metric in the pair. Color and line-type correspond to different strain types. The solid black lines on the plot are reference lines at a value of 1 for no effect, and at 4 and 1/4, effect values which would represent a clinically notable deviation in HAI predictions beyond what is expected from measurement error. Lines represent the mean of the posterior distribution of the contrast and the colored ribbons represent the 95% highest density credible interval (HDCI) for each strain type in each subplot.
(TIF)

**S6 Fig. Pairwise comparisons of predictions (from the GAMMs) between each unique set of two metrics.** The y-axis shows the fold change in predictive titers between metrics, and the two metrics being compared in each subplot are shown as the subplot labels. Each line represents the predictions for the first metric in the pair at a given antigenic distance value divided by the predictions for the second metric in the pair. Color and line-type correspond to different strain types. The solid black lines on the plot are reference lines at a value of 1 for no effect, and at 4 and 1/4, effect values which would represent a clinically notable deviation in HAI predictions beyond what is expected from measurement error. Lines represent the mean of the posterior distribution of the contrast and the colored ribbons represent the 95% highest density credible interval (HDCI) for each strain type in each subplot.
(TIF)

**S7 Fig. Model predictions for both the GAMM and LMM, conditional on the vaccine strain rather than only on the subtype (shown in the main text).** Solid green lines and green ribbons show the mean and 95% highest density continuous interval (HDCI) for GAMM predictions. Dashed orange lines and orange ribbons show the mean and 95% HDCI for LMM predictions. Circular points show the data values. Each subplot shows the model predictions for a particular subtype (changes by row) and distance metric (changes by column). Outcomes shown on the plot are predicted post-vaccination titers for an average individual to an average strain.
(TIF)

**S1 Text. Expanded methods and details on sensitivity analyses.**
(DOCX)

## Acknowledgments

We thank William Michael Landau (Eli Lilly and Company, Indianapolis, IN, USA) and Eric R. Scott (University of Arizona, Tucson, AZ, USA) for their generous help with computational issues and pipeline development. Additionally, we thank Michael A. Carlock (Cleveland Clinic Florida Research & Innovation Center, Port St. Lucie, FL, USA) for assistance with obtaining data. This study was supported in part by resources and technical expertise from the Georgia Advanced Computing Resource Center, a partnership between the University of Georgia's Office of the Vice President for Research and Office of the Vice President for Information Technology.

We gratefully acknowledge all data contributors, i.e., the Authors and their Originating laboratories responsible for obtaining the specimens, and their Submitting laboratories for generating the genetic sequence and metadata and sharing via the GISAID Initiative, on which this research is based. Similarly, we gratefully acknowledge all data contributors for generating the genetic sequence and metadata and sharing via NCBI GenBank and UniProt, on which this research is based.

## Author contributions

**Conceptualization:** W. Zane Billings, Andreas Handel.

**Data curation:** W. Zane Billings, Yang Ge, Amanda L. Skarlupka, Ted M. Ross.

**Formal analysis:** W. Zane Billings.

**Investigation:** Ted M. Ross.

**Methodology:** W. Zane Billings, Yang Ge, Amanda L. Skarlupka, Savannah L. Miller, Hayley Hemme, Murphy John, Andreas Handel.

**Resources:** Ted M. Ross.

**Software:** W. Zane Billings.

**Supervision:** Andreas Handel.

**Validation:** Savannah L. Miller, Hayley Hemme, Murphy John.

**Visualization:** W. Zane Billings.

**Writing – original draft:** W. Zane Billings, Andreas Handel.

**Writing – review & editing:** W. Zane Billings, Yang Ge, Amanda L. Skarlupka, Savannah L. Miller, Hayley Hemme, Murphy John, Natalie E. Dean, Sarah Cobey, Benjamin J. Cowling, Ye Shen, Ted M. Ross, Andreas Handel.

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
