## [Decision Letter · Decision Letter 0]

21 Sep 2025

PCOMPBIOL-D-25-01364

Different antigenic distance metrics generate similar predictions of influenza vaccine response breadth despite moderate correlation

PLOS Computational Biology

Dear Dr. Billings,

Thank you for submitting your manuscript to PLOS Computational Biology. After careful consideration, we feel that it has merit but does not fully meet PLOS Computational Biology's publication criteria as it currently stands. Therefore, we invite you to submit a revised version of the manuscript that addresses the points raised during the review process.

Please submit your revised manuscript within 30 days Nov 21 2025 11:59PM. If you will need more time than this to complete your revisions, please reply to this message or contact the journal office at ploscompbiol@plos.org. Please include the following items when submitting your revised manuscript:

We look forward to receiving your revised manuscript.

Kind regards,

Roger Dimitri Kouyos

Section Editor

PLOS Computational Biology

Roger Kouyos

Section Editor

PLOS Computational Biology

**Journal Requirements:**

6) Please amend your current Competing Interest statement in the online submission form. Please declare all competing interests beginning with the statement "I have read the journal's policy and the authors of this manuscript have the following competing interests:"

**Reviewers' comments:**

Reviewer's Responses to Questions

Reviewer #1: This manuscript describes comparison of several different measures of antigenic distance between influenza vaccine and assay strains and assesses their ability to predict vaccine response. The authors find that even though the different measures are only slightly correlated, they have similar predictive power when it comes to vaccine efficacy. I enjoyed reading this manuscript and particularly appreciate the very thorough descriptions of the methodology found in the supplement. I have only a few very minor comments:

1. The authors often use "which" when they should use "that". Please check correct use of these.

2. Line 235: "difference" should be "different"

3. Line 246: "outcome" is misspelled.

4. Line 343: AIC and BIC are not defined in the manuscript.

5. Line 370: What do the authors mean by "casually adjusted"?

6. Tables 4 and 5. Is is possible to use a smaller font in the metric column so that these names are not split on two lines?

Reviewer #2: The paper considers an interesting and important question — the evaluation the choice of influenza strains when updating flu vaccines. The antigenic distance between 2 strains is typically measured by antigenic cartography. In this approach groups of ferrets are immunized with each of these 2 strains and the extent to which the serum binds the homologous (immunizing) strain vs the heterologous (other) strain gives us a measure of the antigenic distance between these strains. This is a involved and rather difficult approach and the paper considers the use of other measures for the antigenic distance between two (vaccine) strains and how this affects the breadth of vaccine-induced antibody responses the vaccine elicits. They use 3 metrics for antigenic distance between 2 strains: Grantham, p-Epitope and Temporal.

They show many interesting results including the following.

1. Table 2 and Figure 1. There is low ICC across the different antigenic distances for responses to H1 and B lineages and better correlation only for the responses to H3(N2). They might want to discuss why?

2. Figure 2. They then consider how the “breadth of the response” (measured by the post-vaccination titer to the homologous vs heterologous strains) depends on antigenic distance for the different strains and the different measures of antigenic distance.

3. Figure 3. They show that using different antigenic metrics yields similar results for the breadth of the response.

This suggests that the simpler measures for antigenic distance might provide comparable results to antigenic cartography. This is an important and potentially useful result.

I have a couple of minor questions:

1. Can some combination of the different measures of antigenic distance provide improved prediction?

2. How does prior infection history affects the results. This is clearly a question for subsequent studies but might be mentioned in the discussion.

Overall I enjoyed reading the paper and believe it will be a nice addition to the literature.

**Have the authors made all data and (if applicable) computational code underlying the findings in their manuscript fully available?**

Reviewer #1: Yes

Reviewer #2: Yes

PLOS authors have the option to publish the peer review history of their article (what does this mean? ). If published, this will include your full peer review and any attached files.

**Do you want your identity to be public for this peer review?** For information about this choice, including consent withdrawal, please see our Privacy Policy .

Reviewer #1: No

Reviewer #2: No

**Figure resubmission:**
---

## [Decision Letter · Decision Letter 1]

7 Nov 2025

Dear Dr Billings,

We are pleased to inform you that your manuscript 'Different antigenic distance metrics generate similar predictions of influenza vaccine response breadth despite moderate correlation' has been provisionally accepted for publication in PLOS Computational Biology.

Best regards,

Roger Dimitri Kouyos

Section Editor

PLOS Computational Biology

Roger Kouyos

Section Editor

PLOS Computational Biology

Reviewer's Responses to Questions

**Comments to the Authors:**

Reviewer #1: The authors have addressed my previous comments to my satisfaction.

**Have the authors made all data and (if applicable) computational code underlying the findings in their manuscript fully available?**

Reviewer #1: None

PLOS authors have the option to publish the peer review history of their article (what does this mean? ). If published, this will include your full peer review and any attached files.

**Do you want your identity to be public for this peer review?** For information about this choice, including consent withdrawal, please see our Privacy Policy .

Reviewer #1: No

---

## [Editor Report · Acceptance letter]

PCOMPBIOL-D-25-01364R1

Different antigenic distance metrics generate similar predictions of influenza vaccine response breadth despite moderate correlation

Dear Dr Billings,

I am pleased to inform you that your manuscript has been formally accepted for publication in PLOS Computational Biology. Your manuscript is now with our production department and you will be notified of the publication date in due course.

With kind regards,

Anita Estes
